# The role of the Big Two in socially responsible behavior during the COVID-19 pandemic: Agency and communion in adolescents' personal norm and behavioral adherence to instituted measures

**Selma Korlat**[1]*, **Julia Holzer**[1], **Julia Reiter**[1], **Elisabeth Rosa Pelikan**[1], **Barbara Schober**[1], **Christiane Spiel**[1], **Marko Lüftenegger**[1,2]

1 Faculty of Psychology, Department of Developmental and Educational Psychology, University of Vienna, Vienna, Austria, 2 Department for Teacher Education, Centre for Teacher Education, University of Vienna, Vienna, Austria

* selma.korlat@univie.ac.at

**Data Availability Statement:** The data that supports the findings of this study are available at

## Abstract

The outbreak of the COVID-19 virus urged all members of the society to adopt COVID-responsible behavioral patterns and practice them in everyday life. Given the variability in its adoption, it is critical to understand psychological factors associated with socially responsible behavior during the pandemic. This might be even more important among adolescents, who are less endangered by the virus but contribute to its spread. In this article, we focus on adolescent boys' and girls' agency and communion orientations to explain the level of importance they attribute to the instituted measures to contain the spread of the COVID-19 virus (personal norm), as well as their behavioral adherence to those measures. In total, 12,552 adolescents (67.6% girls, $M_{age}$ = 15.06, $SD_{age}$ = 2.44, age range 10–21) answered inventory assessing adolescents' agentic and communal orientation (GRI-JUG) and items related to personal norm regarding the instituted measures and behavioral adherence to the measures. The results showed a small positive role of communion in both boys' and girls' personal norm and behavioral adherence, whereas agency played a very small negative role in boys' and girls' personal norm and boys' behavioral adherence to measures. Nevertheless, these findings could indicate the importance of enhancing communal traits and behaviors in both genders in order to assure socially responsible behavior during the pandemic.

## Introduction

Until the immunization against COVID-19 is fully reached in the population, recommended protective measures remain the main means against the spread of the virus that has already taken more than six million lives and affected many more [1]. The pandemic outbreak forced individuals to change and adapt their behavior by regularly practicing COVID-responsible

the OSF on the following link: (https://osf.io/2a6vp/
?view_only=
4e6b4cf6c444418392ded9748c8f4339).

**Funding:** This work was funded by the Vienna
Science and Technology Fund (WWTF) [https://
www.wwtf.at/] and the MEGA Bildungsstiftung
[https://www.megabildung.at/] through project
COV20-025. BS is the grant recipient of COV20-
025. Open access funding is provided by University
of Vienna. The funders had no role in study design,
data collection and analysis, decision to publish, or
preparation of the manuscript.

**Competing interests:** The authors have declared
that no competing interests exist.

behavior such as physical distancing, regular hand washing and wearing protective gear such
as facemasks. Thus, people were urged to act in a way that is not only beneficial to but pivotal
for their own health, as well as the health of others. Despite the importance of these prosocial
and self-protective behaviors during the pandemic, the public has often been confronted with
the reality that some individuals are not respecting the instituted measures [2]. This variability
in people's adoption of the safety measures and preventive behavior recommended by govern-
ments and public health organizations [3] directly affects the course and severity of the pan-
demic, hence it is crucial to understand the factors that motivate or inhibit their adoption [4].
Although adolescents were perceived as a low-risk group compared to older members of the
society [5,6], they have been seen as potential transmitters of the virus, thus contributing to its
spread [7]. Therefore, scholars emphasize the importance of psychological factors contributing
to variation in adolescents' adoption of the instituted measures [8]. In this article, we focus on
adolescent boys' and girls' agency and communion orientations to explain the level of impor-
tance they attribute to the instituted measures to contain the spread of COVID-19 –their per-
sonal norm, as well as their behavioral adherence to those measures.

## Agency and communion in prosocial and socially responsible behavior

Agency and communion reflect fundamental human traits and are seen as two pillars of
human personality [9], also known as the "Big Two" [10]. Agency is the positive value placed
on individuality, personal striving and self-achievement. Communion is the positive value
placed on relationships and benefiting others, even the society as a whole [11–13]. Agency, or
the "getting ahead" trait, includes qualities such as "assertive", "independent", and "ego-cen-
tric". Communion, or the "getting along" trait, depicts characteristics such as "cooperative",
"empathic", and "trustworthy" [11,12,14]. Research has shown that people who ascribe more
communal qualities to themselves have stronger communal motives, while people who ascribe
more agentic qualities to themselves have stronger agentic motives [15]. Communal motives
involve helping others in terms of providing emotional or material support, as well as acting
for the benefit of others [16]. Communal collaboration and altruism may involve connecting
not only with proximal others, but with distal others, such as society or a broader community
[17]. In general, communal qualities could be perceived as more moralistic or prosocial, while
agentic ones could be perceived as more egocentric in nature [10,13].

Individual differences in the ways people pursue agentic and communal motives define not
only their self-description and perception of others, but their values, attitudes and behavior
[12,18]. According to Bakan's classic perspective [9], a behavior is driven by communion or
agency insofar as aspects of a given behavior are aligned with or allow the expression of com-
munion or agency. For instance, communion has been conceptualized as the Big Two person-
ality basis of prosocial behavior–voluntary behavior intended to benefit others [e.g., [19,20]–as
that kind of behavior affords the expression of high communion [9,21]. Agency, on the other
hand, has been conceptualized as unrelated to prosociality, given that this behavior rarely
affords the expression of agency and sometimes even inhibits it [21,22]. The positive role of
communion in prosocial behavior has been confirmed in a previous large cross-country study
with adults [22]. A study with early adolescents also supported Bakan's expressiveness perspec-
tive [9], showing communal goals as positive predictors of prosocial behavior. Agentic goals
were a significant negative predictor in this study [23]. A recent study showed that adolescents'
prosocial tendencies fall into two domains–serving others and self-serving propensities, argu-
ing that adolescents can act prosocially with the intention to help others or to gain personal
benefit [24]. Moreover, this study showed that this differentiation in prosociality is supported
through patterned associations with empathic versus egoistic individual differences, taking

into account empathy as an other-directed factor, and approval needs and narcissism as self-oriented personal characteristics.

Socially responsible behavior during the pandemic encompasses both domains of prosocial behavior. On one hand, one could place high importance on the instituted measures and adhere to them due to the underlying motive to help and protect others, for instance persons in risk groups, family or older members of a society. On the other hand, the motivation for such behavior during the pandemic can also be self-serving–driven by the intention to protect oneself. A recent study showed that focusing on one's community promotes increased intentions to wear a face covering [25], suggesting the importance of other-oriented motives for socially responsible behavior. Moreover, Zajenowski et al. [26] found agreeableness, among the Big Five traits, to predict compliance of COVID instituted measures, arguing that this might be due to higher compassion and caring toward others that characterize agreeable people. At the same time, scholars argue that the threat to life and disrupted daily routine decreased one's agency, which might have generated a specific tension and led to changes in the direction of agency during the pandemic, resulting in an increase in self-oriented motives as a compensatory mechanism [27]. For instance, a study by Leder et al. [28] indicated that participants mostly followed guidelines that protected themselves rather than the general public. Thus, the COVID-19 situation could allow the expression of both agency and communion, as both dimensions might play a positive role in personal norm–perceived importance of and behavioral adherence to the instituted measures.

## Gender roles and prosocial behavior

Kindness, caring for others and other communal qualities have traditionally been associated with women more than men, while goal and achievement orientation and agentic qualities have been associated with men more than women in Western societies [11,29]. These differentiations reflect gender stereotypes, according to which women's gender role is to be caring, gentle, concerned with others, and to get along. Men's gender role, on the other hand, is to be assertive, dominant, decisive and to get ahead [30,31]. Through socialization processes, gender role beliefs are grafted onto boys' and girls' self-concept and internalized in gender identities, acting as personal dispositions [32,33]. Consequently, studies repeatedly show that, adolescent girls report a higher communal self-concept and endorse more communal goals than adolescent boys, whereas boys report a relatively agentic self-concept and pursue more agentic goals than girls [23,34,35]. Hence, one would expect socially responsible behavior in girls and women more than in boys and men.

However, social-role theory posits that gender differences in prosocial behavior depend on gender role acquired during socialization: male gender role fosters helping that is agentic (e.g., heroic and chivalrous), whereas the female gender role fosters helping that is communal (e.g., nurturant and caring) [36]. Indeed, it has been shown that males are more prosocial in situations where helping behaviors drew on agency, such as dangerous emergencies, accidents that encounter strangers, chivalrous help to women, and collectivist support to families, organizations, and nations at war [37]. Females, on the other hand, are more likely to help and provide emotional support in close relationships, for example to spouses and friends [38], elderly relatives [37], and to engage in prosocial citizenship behaviors [e.g., altruism; 39,40].

Previous studies investigating behaviors related to serving others (such as consoling, helping, and sharing with their peers) in adolescence showed dominance of girls over boys in prosociality [23,41–43]. These studies did not investigate the role of agency and communion for such behaviors separately in boys and girls. However, one might expect patterns of prosocial adolescents' behavior to be in line with gender roles–as suggested by the social-role theory, just

intensified, as gender-specific socialization pressures are assumed to strengthen boys' and girls' adherence to gender roles and gender stereotypes in this period of life [44].

## Gender differences in socially responsible behavior during the COVID-19 pandemic

A recent study on gender differences in the context of COVID-responsible behavior in Western Europe, Australia and the US with an adult sample showed strong gender differences favoring women over men in both their agreement and compliance with public health rules [45]. Other studies from China and Iran point in the same direction, arguing that a reason for these gender differences might be women's greater responsibility and more significant concern about infecting others [46,46,47].

A study with adolescent samples showed a similar pattern of gender differences with girls reporting greater social distancing compared to boys [8]. The same authors found a negative relationship between self-interest values (which might be agentic in nature) and social distancing behavior, while social responsibility (which might align with communal orientation) was not a statistically significant predictor of social distancing, but was positively related to disinfecting among adolescents in the US. Other studies found the same pattern of gender differences in Norway [48] and Canada [7]. The Canadian study found that self-perceived risk, as well as the motive to help flatten the disease curve or to protect significant others were positive factors for adherence to preventive measures among youth in Canada. However, none of these studies investigated gender differences with respect to agency and communion orientation in COVID-responsible norm and behavior.

## Present study

The central goal of this study is to test Bakan's classic perspective [9] in the situation of socially responsible behavior during the COVID-19 pandemic, applying postulates of social-role theory on gender differences in such behavior. In order to do so, we investigated boys' and girls' agency and communion orientations to explain their personal norm regarding the instituted measures to contain the spread of COVID-19, as well as their behavioral adherence to those measures.

As both agency and communion might play a significant positive role in socially responsible behavior during the COVID-19 pandemic, we propose that the present situation affords the expression of both agency and communion. Hence, we expect both agency and communion to positively predict personal norm and behavioral adherence to instituted measures during the COVID-19 pandemic. However, as social-role theory [37] posits a different influence of agency and communion in prosocial behavior of males and females due to their respective gender roles, we propose that communion will be a stronger positive predictor of personal norm and behavior among girls than boys, while agency will be a stronger positive predictor of personal norm and behavior among boys than girls. As previous studies indicated changes in levels of helping and prosocial behaviors throughout adolescence [e.g., 43,49,50], we controlled for age in the analyses.

## Materials and methods

### Participants, procedure and context of data collection

The data were collected from November 22 until December 6, 2020 in Austria, as part of a larger project investigating learning under the conditions of the COVID-19 (redacted for review). For the purposes of this study, a subsample consisting of boys and girls only was

selected, excluding 0.6% of students that declared their gender as diverse and 2.1% of those who did not want to share information on their gender. In total, the selected study sample comprised 12,552 adolescents (67.6% girls, $M_{age}$ = 15.06, $SD_{age}$ = 2.44, age range 10–21). Data were collected with online questionnaires. To recruit participants, we distributed the link to the online questionnaire by contacting manifold stakeholders such as school boards, educational networks, and school principals with the help of the Austrian Federal Ministry for Education, Science, and Research. Sample size was left open, with the goal to collect as many participants as possible within the pre-determined data collection period. Participation was voluntarily and anonymous. Only participants who gave active consent were included in the dataset. During the data collection, Austria was in a second total lockdown. Throughout the entire data collection period, a stay-at-home order was in effect. Citizens were asked not to leave home except in situation of danger, familial responsibilities (such as caring for sick or old relatives), purchasing essential items (food, medicine, etc.) and physical and mental relaxation (such as taking walks). Home-schooling was implemented for students of all ages. In case of leaving home for stated reasoning, social distancing measures (masks, minimum 1.5 meters distance, regular hand washing and disinfection) were strongly recommended [51].

## Measures

**Agency and Communion** were assessed using 10 positive traits from the Inventory for Measuring Adolescents' Gender Role Self-concept [GRI-JUG] in adolescents [52]. Examples of agentic attributes were "courageous" and "strong". Examples of communal attributes were "empathic" and "emotional". Participants were asked to rate to what extent each attribute applies to them on a scale from 1 –applies [almost] always to 5 –does not apply at all. Analyses were conducted with recoded items so that higher values reflected higher agreement with the statements.

 **Personal norm and behavioral adherence to COVID measures.** In line with other studies investigating the issue [e.g., 53,54], personal norm–perceived importance of the instituted measures to contain the spread of COVID-19 –was measured using the following item: "How important do you find adhering to the Corona-rules?" [ranging from 1 –very important to 5 – not important at all]. To measure their behavioral adherence to those measures, participants were asked: "To what extent do you adhere to the Corona-rules?" [ranging from 1 –totally to 5 –not at all]. Analyses were conducted with recoded items so that higher values reflected higher agreement with the statements.

## Analyses

Data were analyzed using SPSS version 25.0 and Mplus version 8.4 [55]. All statistical significance testing was performed at the .05 level. However, due to the large sample, rather than relying on statistical significance, we particularly focused on the identified effect sizes for the regression parameters when interpreting the results. In doing so, we followed Gignac & Szodorai [56] guidelines, according to which standardized values of .10, .20, and .30 reflect small, moderate, and large effect sizes, respectively. Following the ASA guideline on p-values [57], significant p-values are supported with 95% two-sided confidence intervals in all tables in the results section.

 First, CFA using principal component analysis was conducted to analyze the construct validity of agency and communion scales. Goodness-of-fit was evaluated using $\chi^2$ test of model fit, CFI, TLI, RMSEA and SRMR. We considered typical cutoff scores reflecting excellent and adequate fit to the data, respectively: (a) CFI and TLI > 0.95 and 0.90; (b) RMSEA and SRMR < .06 and .08 [58].

Second, we set up two models to test main effects and moderation by gender. To test the role of agency and communion in socially responsible behavior during the COVID-19 pandemic (H1), a structural equation modeling (SEM) was conducted with agency and communion as predictors, and age as a control variable. Personal norm and behavioral adherence to the measures served as dependent variables. To test the role of agency and communion in girls (H2) and boys (H3), the same model was conducted as a multigroup analysis, with gender as moderator defining models separately for boys and girls. We also used the Mplus MODEL CONSTRAINT command to test the difference in regression slopes for the direct effects for girls vs. boys for statistical significance.

## Results

### Preliminary analyses

Operationalizing agency and communion with five indicators resulted in poor model fit, $\chi^2(34) = 2899.156$, $p < .001$, CFI = .845, TLI = .794, RMSEA = .082, SRMR = .074. In order to reduce indicators, we conducted first an exploratory factor analysis with maximum likelihood estimation with the aim of identifying the items with the highest factor loadings. The rotation converged in three iterations (see S1 Table in the supplementary material). Accordingly, the attributes "romantic" and "industrious" were removed from the communion scale, as they had both the lowest communalities within the scale and the smallest factor loadings in both the unrotated and rotated version. Following the same reasoning, the items "companionable" and "humorous" were removed from the agency scale. A follow up confirmatory principal component analysis yielded higher factor loadings for the remained items (see S2 Table in the supplementary material) and better model fit, $\chi^2(8) = 156.003$, $p < .001$, CFI = .987, TLI = .975, RMSEA = .038, SRMR = .021. Thus, to measure agency and communion, three agentic attributes (courageous, sporty, and strong; composite reliability—CR = .836) and three communal attributes (emotional, sympathetic and empathic; CR = .838) were used. Measurement invariance testing was conducted using MPlus to ensure that the agency and communion items were indicators for the same latent constructs in both male and female adolescents. CFI and RMSEA were used as indicators for absolute goodness of model fit. Relative model fit was assessed by comparing BICs of the nested models, with smaller BIC values indicating a better trade-off between model fit and model complexity (van de Schoot et al., 2012). Table 1 displays the model fit information for the three levels of measurement invariance (configural, metric, and scalar) tested for both agency and communion. For both agency and communion, relative model fit was best (i.e., the BIC was lowest) for the metric model. The absolute model fit was

**Table 1. Measurement invariance testing across groups for the confirmatory factor analytic measurement models for agency and communion.**

| Model | $\chi^2$ | df | CFI | RMSEA | BIC | Model description |
|---|---|---|---|---|---|---|
| *Agency* | | | | | | |
| | 0.000* | 0 | 1.000 | 0.000 | 105773.103 | Configural invariance |
| | 12.762* | 2 | 0.998 | 0.029 | 105768.569 | Metric invariance |
| | 60.130* | 4 | 0.989 | 0.047 | 105800.354 | Scalar invariance |
| *Communion* | | | | | | |
| | 0.001* | 0 | 1.000 | 0.000 | 87530.072 | Configural invariance |
| | 2.779 | 2 | 1.000 | 0.008 | 87515.266 | Metric invariance |
| | 34.862* | 4 | 0.992 | 0.035 | 87536.498 | Scalar invariance |

*p < .001.

**Table 2. Descriptive statistics and bivariate correlations for all variables used in the SEM model, calculated for the complete sample (male and female).**

|  | M [SD] | 1 | 2 | 3 | 4 | 5 | 6 | 7 | 8 | 9 | 10 |
|---|---|---|---|---|---|---|---|---|---|---|---|
| [1] Gender | 1.32 [0.47] |  |  |  |  |  |  |  |  |  |  |
| [2] Age | 15.06 [2.44] | -.14** |  |  |  |  |  |  |  |  |  |
| [3] Norm | 4.34 0[.87] | -.04** | -.07** |  |  |  |  |  |  |  |  |
| [4] Behaviour | 4.42 [0.75] | -.02* | -.06** | .67** |  |  |  |  |  |  |  |
| [5] C1 [emotional] | 4.20 [0.89] | -.20** | -.03** | .08** | .09** |  |  |  |  |  |  |
| [6] C4 [sympathetic] | 4.34 [0.78] | -.14** | .04** | .10** | .12** | .35** |  |  |  |  |  |
| [7] C5 [empathetic] | 4.13 [0.95] | -.19** | .04** | .08** | .08** | .54** | .46** |  |  |  |  |
| [8] A3 [courageous] | 3.67 [1.01] | .10** | -.17** | -.03** | -.02 | .02* | .12** | .05** |  |  |  |
| [9] A4[sporty] | 3.59 [1.25] | .14** | -.22** | -.00 | .01 | .08** | .08** | .08** | .39** |  |  |
| [10] A5 [strong] | 3.72 [1.04] | .12** | -.14** | -.00 | .02* | .03** | .08** | .07** | .54** | .41** |  |

All scales range from 1 indicating the lowest value to 5 indicating the highest value.

** p < .01.

* p < .05.

excellent for these models for both latent constructs. Hence, the results are interpreted under the assumption of metric measurement invariance: The assumption that the factor structure of the latent constructs agency and communion was the same for both boys and girls, and the factor loadings were similar between both groups as well. Table 2 provides bivariate correlations among all variables as well as descriptive statistics for the entire sample.

## Main analyses

The SEM model 1 testing H1 showed a good model fit ($\chi^2$ (22) = 1094.29, CFI = 0.954, RMSEA = 0.062). As presented in Fig 1, communion positively predicted both personal norm ($\beta$ = 0.122, SE = 0.010, $p$ < .001, CI (0.105, 0.139)) and behavioral adherence to the measures ($\beta$ = 0.129, SE = 0.010, $p$ < .001, CI (0.112, 0.146)). Agency was a small negative predictor of personal norm ($\beta$ = -0.049, SE = 0.011, $p$ < .001, CI (-0.066, -0.031)) and behavioral adherence to the measures ($\beta$ = -0.026, SE = 0.011, $p$ = .016, CI (-0.043, -0.008)). Age was weakly negatively correlated to both outcome variables (personal norm $\beta$ = -0.078, SE = 0.009, $p$ < .001, CI (-0.093, -0.064); behavioral adherence $\beta$ = -0.070, SE = 0.009, $p$ < .001, CI (-0.085, -0.055)).

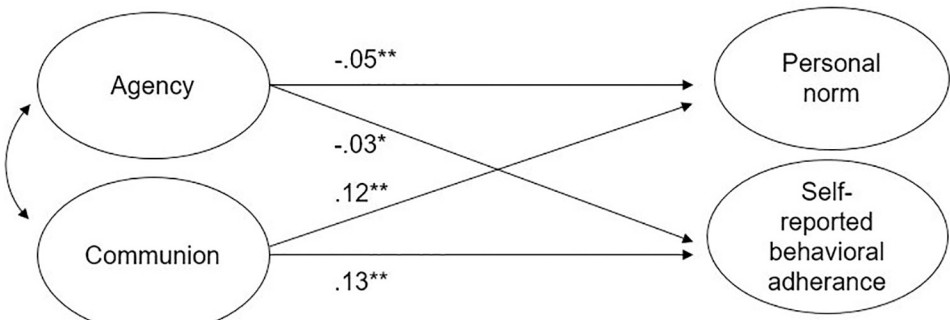

**Fig 1. Structural equation model predicting personal norm and self-reported behavioral adherence to COVID measures (Model 1).** This SEM predicts personal norm and self-reported behavioral adherence to COVID measures from adolescents' agentic and communal orientation. Statistics are standardized regression coefficients. *$p$ < .05 **$p$ < .001.

The two-group SEM (model 2) exhibited a good level of fit to the data ($\chi^2$(52) = 1208.94, CFI = 0.949, RMSEA = 0.060). For girls, communion positively predicted both personal norm regarding social responsible behavior ($\beta$ = 0.122, SE = 0.013, $p < .001$, CI (0.102, 0.143)) and behavioral adherence to the measures during the COVID-19 pandemic ($\beta$ = 0.138, SE = 0.013, $p < .001$, CI (0.117, 0.159)), while agency was a small statistically significant negative predictor of personal norm only ($\beta$ = -0.030, SE = 0.013, $p$ = .022, CI (-0.052, -0.008)). No statistically significant effect of agency was found for girls' behavioral adherence to COVID measures ($\beta$ = 0.005, SE = 0.013, $p$ = .681, CI (-0.016, 0.027)). For boys, communion was also a positive predictor of both personal norm ($\beta$ = 0.116, SE = 0.019, $p < .001$, CI (0.084, 0.147)) and behavioral adherence to the measures during the COVID pandemic ($\beta$ = 0.121, SE = 0.019, p < .001, CI (0.089, 0.153)). Agency, on the other hand, weakly significantly negatively predicted both personal norm ($\beta$ = -0.077, SE = 0.019, $p < .001$, CI (-0.108, -0.045)) and behavioral adherence to the measures during the pandemic ($\beta$ = -0.085, SE = 0.019, $p < .001$, CI (-0.116, -0.053)). Fig 2 displays coefficients separately for girls and boys.

The difference in regression slopes was statistically significant for agency predicting both outcome variables (personal norm $\beta$ = 0.062, SE = 0.029, $p$ = .034, CI (0.014, 0.111); behavioral adherence $\beta$ = 0.098, SE = 0.025, $p < .001$, CI (0.057, 0.140)), with a statistically significantly stronger negative relation found for boys.

## Discussion

The main goal of this study was to investigate the role of agency and communion in boys' and girls' personal norm regarding the importance of the instituted measures to contain the spread of the COVID-19 virus, as well as their behavioral adherence to those measures. Although it was assumed that in the pandemic context both agency and communion might be positively

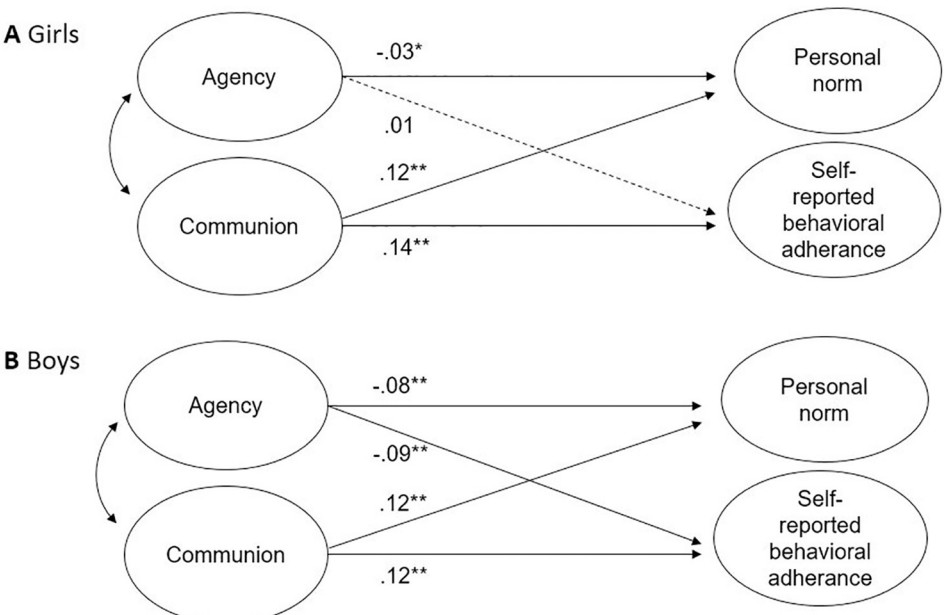

**Fig 2. Structural equation model predicting personal norm and self-reported behavioral adherence to COVID measures in male and female sub-sample (Model 2).** This SEM predicts personal norm and self-reported behavioral adherence to COVID measures from girls' (panel A) and boys' (panel B) agentic and communal orientation. Statistics are standardized regression coefficients. Dotted lines represent non-significant relations. $^*p < .05$ $^{**}p < .001$.

related to personal norms regarding the importance of the COVID measures and behavioral adherence to them, our results showed only communion as a positive predictor and agency as a negative predictor of those aspects of socially responsible behavior among adolescents. However, the effect sizes for both agency and communion were minor to small, and even smaller for agency than for communion. In general, this aligns with a previous study investigating the role of personality traits in compliance to the COVID measures [26], which found that the way people perceived the situation explains more variance in compliance than personality traits. The authors argue that this finding supports the "strong situation hypothesis" according to which personality traits have less room to play an important role in predicting behaviors when situational cues overpower dispositional tendencies [26], which might be the case in the current pandemic context.

Although the small effect sizes limit the practical relevance of the identified coefficients, they indicate potentially higher importance of communion than agency aligning with Bakan's classic perspective [9] of prosocial behavior according to which prosociality can afford expression of communion, but not expression of agency. Similarly, this finding is, to some extent, in line with studies on prosocial behavior among adolescents [23], as well as among adults [22] that showed communion as a positive, but agency as a negative factor. Moreover, Gebauer et al. [22] found communion to be a stronger predictor of prosocial behavior in countries where interest in prosociality is common, and weaker in countries where such behavior is uncommon (*communal social assimilation*). In contrast, agency negatively predicted interest in prosocial behavior comparatively weakly in countries where interest in prosociality was common, and comparatively strongly in sociocultural contexts where such interest was uncommon (*agentic social contrast*). Applying that to our results, and insofar prosocial behavior can be equated to socially responsible behavior during the pandemic, this could indicate a stronger interest in prosociality in Austria where data was collected.

Our finding also aligns with Zajenowski et al. [26] and Capraro & Barcelo [25] who showed that qualities and behavior related to others (orientation to one's community and being agreeable) are related to higher compliance with measures during COVID-19. It could be that, overall, socially responsible behavior during the COVID-19 pandemic is driven by the motivation to protect others more than oneself, thus supporting the expression of communion, but not agency. This is especially plausible in this age group given that adolescents were less endangered by COVID-19 virus than older persons [5,6]. It might be that communal adolescents with high levels of empathy and sympathy for others that were at higher risk perceived COVID measures as more important and, consequently, complied with them. Thus, being oriented toward others might foster social responsible behavior during the pandemic among youth, while self-orientation focus might impede it. The latter result aligns with Oosterhoff & Palmer's [8] finding on negative relationship between self-interest values and social distancing behavior. It might be that agentic orientation in adolescents exceeds the focus on self-protection from the virus, and encompasses broader self-orientation (e.g., putting one's own needs before the needs of others) that might hinder socially responsible behavior.

Even when tested separately for boys and girls, communion was found to positively predict both aspects of socially responsible behavior in both groups. Albeit small in general, effect sizes were slightly larger in the girls' than in the boys' sub-sample. Nevertheless, results showed insignificant differences in slopes for communion predicting social responsible behavior, indicating equal importance of communion in socially responsible behavior among adolescents across genders. Agency, on the other hand, weakly negatively predicted both aspects of socially responsible behaviors in boys and perceived importance in girls. Moreover, results showed differences in slopes for agency predicting both observed aspects of social responsible behavior during the pandemic, indicating stronger negative relation in boys' sub-sample than in girls'

sub-sample. This is consistent with our hypothesis that assumed stronger effects of agency in boys, but the direction of that relationship is contrary to our hypothesis: While stronger positive effects of agency were expected in boys, stronger negative ones emerged in the results. However, effect sizes for agency also in moderated model were minor, indicating low predictive power of agency in the present study. While one reason for that could be lack of content validity of items used to measure agency, another explanation might be a truly negligible role of agentic orientation and universal positive role of communion in socially responsible behavior during the pandemic beyond the effects of gender in adolescence. This assumption is in line with higher increase in communion than in agency during the pandemic as human *behavioral immune mechanism* under threat from an infectious disease [27]. Previous studies have found that presence of infectious disease leads to increase of collectivism [59] and in-group favoritism [60]–other-oriented motives. Taking that into account, consistent gender differences in terms of socially responsible behavior during COVID-19 (with women consistently behaving more responsibly) reported in other studies [e.g., 45–47] might be related to higher communion in women. Future studies are needed to clarify the role of agency and communion in socially responsible behavior in adults.

In sum, our findings indicate a small but positive role of communion for socially responsible behavior in relation to COVID safety measures in both gender groups during adolescence. These findings might have practical implications for communication patterns or pedagogical practices for parents and teachers. Although previous studies showed parental and peers' disapproval of boys' deviation from agency toward communion [61], our study indicates that enhancing communal traits and behaviors in both boys and girls might contribute, although to small extent, to socially responsible behavior during the COVID pandemic. Endorsement of valuing relationships and benefiting others (the "getting along" motive) in boys could potentially increase their prosociality and helping behavior that draws on communion in general. At the same time, it might decrease the negative impact of pressures young boys face to prove their masculinity and avoid appearing communal at any cost [e.g., 62].

## Limitations and future directions

While this study has several strengths, including a large sample size, some limitations must be considered. First, the results rely on self-reports and analyses are correlational in nature, limiting causal inference. Data were collected online, which led to a self-selection of the sample and as a consequence to an overrepresentation of girls [63] and potentially persons with general communal orientation. Moreover, due to the lack of theoretical argumentation and consequent difficulties to pose grounded hypotheses, 0.6% of our sample who declared their gender as diverse had to be deleted from analysis. The theories on gender role orientations should be revised and broadened in order to include non-binary individuals. Second, low predictive power of agency and communion in the present study could indicate low content validity of the items used in the present study, as well as other confounding variables (such as health anxiety) at play. Future studies should include other agency and communion measures focusing especially on different facets of agency and communion in socially responsible and prosocial behavior [64], as well as other related constructs. Furthermore, future studies investigating the role of agency and communion in this context might include more elaborative items on socially responsible behavior during and beyond the COVID-19 pandemic. Third, the adherence to instituted measures was assessed with a single, self-report item, measuring people's declarations about their adherence rather than their actual behavior. Future studies might include specific questions about behaviors or track behavior over a shorter period (e.g., within the last week) using, for example, mobile phone applications. Finally, the sample was W.E.I.R.

D. [65], which limits the generalizability of our findings. In societies with more different cultural contexts and different agency-communion orientations, a different interplay between the variables might be observed.

## Supporting information

**S1 Table. Communalities and results of unrotated and rotated maximum likelihood analysis (factor loadings).**
(DOCX)

**S2 Table. Communalities and results of unrotated and Oblimin-rotated principal component analysis (factor loadings).**
(DOCX)

## Acknowledgments

### Ethical statement

Ethical review and approval was not required for the study on human participants in accordance with the local legislation and institutional requirements. Written informed consent for participation was not provided by the participants' legal guardians/next of kin because data was collected online due to circumstances of the COVID-19 pandemic and only consent from the students themselves was collected. The study complies with the research guidelines of the Federal Ministry for Education, Science, and Research who supported the study. Participation in the study was completely voluntarily. Before being forwarded to the items, participants were informed about the study's goals, approximate duration of the questionnaire, inclusion criteria for participation, and the complete anonymity of their data. Only those who gave active consent were included in the dataset. The study was carried out in accordance with the European General Data Protection Regulation.

## Author Contributions

**Conceptualization:** Julia Holzer.

**Data curation:** Selma Korlat, Julia Holzer, Elisabeth Rosa Pelikan, Barbara Schober, Christiane Spiel, Marko Lüftenegger.

**Formal analysis:** Selma Korlat, Julia Reiter.

**Funding acquisition:** Barbara Schober, Christiane Spiel, Marko Lüftenegger.

**Investigation:** Selma Korlat, Elisabeth Rosa Pelikan, Barbara Schober, Christiane Spiel, Marko Lüftenegger.

**Methodology:** Selma Korlat, Julia Holzer, Julia Reiter, Marko Lüftenegger.

**Project administration:** Selma Korlat.

**Visualization:** Selma Korlat.

**Writing – original draft:** Selma Korlat.

**Writing – review & editing:** Julia Holzer, Julia Reiter, Elisabeth Rosa Pelikan, Barbara Schober, Christiane Spiel, Marko Lüftenegger.

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
