## [Decision Letter · Decision Letter 0]

2 Apr 2022

PONE-D-22-00800The Role of the Big Two in Socially Responsible Behavior During COVID-19: Agency and Communion in Adolescents’ Personal Norm and Behavioral Adherence to Instituted MeasuresPLOS ONE

Dear Dr. Korlat,

Thank you for submitting your manuscript to PLOS ONE. After careful consideration, we feel that it has merit but does not fully meet PLOS ONE’s publication criteria as it currently stands. Therefore, we invite you to submit a revised version of the manuscript that addresses the points raised during the review process.  Please submit your revised manuscript by May 17 2022 11:59PM. If you will need more time than this to complete your revisions, please reply to this message or contact the journal office at plosone@plos.org. Please include the following items when submitting your revised manuscript:A rebuttal letter that responds to each point raised by the academic editor and reviewer(s). You should upload this letter as a separate file labeled 'Response to Reviewers'.A marked-up copy of your manuscript that highlights changes made to the original version. You should upload this as a separate file labeled 'Revised Manuscript with Track Changes'.An unmarked version of your revised paper without tracked changes. You should upload this as a separate file labeled 'Manuscript'.

We look forward to receiving your revised manuscript.

Kind regards,

Peter Karl Jonason

Academic Editor

PLOS ONE

Journal Requirements:

2. In your ethical statement, you stated that: "Ethical review and approval was not required for the study on human participants in accordance

with the local legislation and institutional requirements...The study complies with the ethical standards of the Federal Ministry for Education, Science, and Research who approved of the study.". As these two statements appear to be contradictory, please confirm whether ethical approval was obtained for this study.

Reviewers' comments:

Reviewer's Responses to Questions

**Comments to the Author**

1. Is the manuscript technically sound, and do the data support the conclusions?

Reviewer #1: Yes

2. Has the statistical analysis been performed appropriately and rigorously? 

Reviewer #1: Yes

3. Have the authors made all data underlying the findings in their manuscript fully available?

Reviewer #1: Yes

4. Is the manuscript presented in an intelligible fashion and written in standard English?

Reviewer #1: Yes

5. Review Comments to the Author

Reviewer #1: 1. I was a bit surprised to read in the Method section how “personal norm” was operationalized. I expected somewhat different from reading the Introduction. Perhaps the authors could clarify early on in the manuscript how precisely they use the term “personal norm” in their research.

2. Good decision to drop some items. They not only evidenced poor psychometric properties, but also poor face validity (IMO).

3. Are the effects of agency and communion unique effects? That is, is the shared variance between agency and communion pulled out statistically? Figure 1 suggests that it is not. Research on the Big Two and norm conformity, however, typically focuses on those unique effects (e.g., Gebauer, Paulhus, & Neberich, 2013, SPPS).

4. The negative relation between agency and norm adherence is described as counter to expectations. However, from the perspective of extant research on the Big Two and conformity to sociocultural norms this negative relation is the expected one (if anything, the very small size of that relation is surprising) (for a review, see Gebauer, Lüdtke, Sedikides, & Neberich, 2014, JOPY)

6. PLOS authors have the option to publish the peer review history of their article (what does this mean?). If published, this will include your full peer review and any attached files.

Reviewer #1: No

---

## [Author Response · Author response to Decision Letter 0]

2 May 2022

Dr. Peter Karl Jonason

Editor, PLOS ONE

MS number: PONE-D-22-00800

Title: The role of the Big Two in socially responsible behavior during the COVID-19 pandemic: Agency and communion in adolescents’ personal norm and behavioral adherence to instituted measures

Dear Dr. Jonason,

Thank you for your response to our submission to the PLOS ONE. We appreciate the valuable review, as well as the opportunity to resubmit the manuscript. Please find below our responses on the set of comments and recommended actions. As suggested, the file Response to reviewers is uploaded separately.

We look forward to hearing from you regarding our revised manuscript and are happy to respond to any further questions and comments you may have.

Sincerely,

The authors

 

PONE-D-22-00800

Below is a list of concerns raised by the journal regarding MS number: PONE-D-22-00800 and the changes we have made in response.

and

The manuscript is revised now so that it meets PLOS ONE's style requirements.

2. In your ethical statement, you stated that: "Ethical review and approval was not required for the study on human participants in accordance with the local legislation and institutional requirements...The study complies with the ethical standards of the Federal Ministry for Education, Science, and Research who approved of the study.". As these two statements appear to be contradictory, please confirm whether ethical approval was obtained for this study.

We apologize for this confusion. The statement is corrected now so that the applied ethical procedure is clear (page 22):

“Ethical review and approval was not required for the study on human participants in accordance with the local legislation and institutional requirements. Written informed consent for participation was not provided by the participants’ legal guardians/next of kin because data was collected online due to the circumstances of the COVID-19 pandemic and only consent from the students was collected. The study complies with the research guidelines of the Federal Ministry for Education, Science, and Research who supported the study. Participation in the study was completely voluntarily. Before being forwarded to the items, participants were informed about the study’s goals, approximate duration of the questionnaire, inclusion criteria for participation, and the complete anonymity of their data. Only those who gave active consent were included in the dataset. The study was carried out in accordance with the European General Data Protection Regulation.”

Captions are included at the end of the manuscript now and in-text citations are updated.

Reference list is updated now: 

a) doi numbers have been added to all articles where available (following the proposed format in https://journals.plos.org/plosone/s/file?id=wjVg/PLOSOne_formatting_sample_main_body.pdf)

b) data from reference No. 1 on the list – the number of COVID-19 cases according to WHO – has been updated, as well as the citation date accordingly

c) a mistake in referencing article No. 22 on the list has been corrected

PACE corrected figures are now updated and saved as Fig1.tif and Fig2.tif files.

Additional comment:

Please note: Grammar mistakes spotted in the manuscript have been corrected without track changes.

Reviewer 1

Dear Reviewer,

Thank you very much for your time and valuable review. Your and Editor’s feedback improved our manuscript. Please find below our responses on your comments and recommended actions.

Sincerely,

The authors

 

Reviewer 1

1. I was a bit surprised to read in the Method section how “personal norm” was operationalized. I expected somewhat different from reading the Introduction. Perhaps the authors could clarify early on in the manuscript how precisely they use the term “personal norm” in their research.

Thank you very much for this remark. We indicate throughout the manuscript now how exactly we use the term “personal norm”:

In the introduction (page 3):

“In this article, we focus on adolescent boys’ and girls’ agency and communion orientations to explain the level of importance they attribute to the instituted measures to contain the spread of COVID-19 – their personal norm, as well as their behavioral adherence to those measures.”

In the theoretical part (page 6):

“Thus, the COVID-19 situation could allow the expression of both agency and communion, as both dimensions might play a positive role in personal norm – perceived importance of and behavioral adherence to the instituted measures.”

In the discussion (page 16):

“The main goal of this study was to investigate the role of agency and communion in boys’ and girls’ personal norm regarding the importance of the instituted measures to contain the spread of COVID-19 virus, as well as their behavioral adherence to those measures. Although it was assumed that in the pandemic context both agency and communion might be positively related to personal norms regarding the importance of the COVID measures and behavioral adherence to them, our results showed only communion as a positive predictor and agency as a negative predictor of those aspects of socially responsible behavior among adolescents.”

2. Good decision to drop some items. They not only evidenced poor psychometric properties, but also poor face validity (IMO).

We also agree that the face validity of excluded items was low.

3. Are the effects of agency and communion unique effects? That is, is the shared variance between agency and communion pulled out statistically? Figure 1 suggests that it is not. Research on the Big Two and norm conformity, however, typically focuses on those unique effects (e.g., Gebauer, Paulhus, & Neberich, 2013, SPPS).

Thank you for this question. In our analyses, both agency and communion were included in the models together in order to control for the influence of one variable (e.g., agency) while testing the predictive role of another variable (e.g., communion). This way, our analytic procedure is in line with Gebauer et al. (2013) in which they emphasize the step of controlling for one over the effect of another Big Two dimension. Moreover, the correlation between agency and communion in our sample is small to moderate (r = .289, p <.001), similar to the study of Gebauer et al. (2014; r = .34). To examine potential problems regarding multicollinearity, we computed the VIF for agency and communion, yielding the following results:

VIF (agency) = 2.08

VIF (communion) = 1.49

Generally, VIFs higher than 5 are considered indicative of collinearity, suggesting potential difficulties in separating out the independent contribution of the variables concerned (James et al., 2013). Other authors suggest a more conservative approach, considering VIFs greater than 2.5 indicative of collinearity (Johnston et al., 2018). Even if the stricter criterion is applied, our results suggest absence of any difficulties in separating out the independent contribution of agency and communion, with current results indicating their unique effects.

If there are any additional steps and procedures we can conduct to answer your question and assure adequate analytic procedure, we would be more than happy and eager to implement them. 

P.S.: In line with the result of a significant correlation between our predictors, factor correlation has been specified in our Figures. 

4. The negative relation between agency and norm adherence is described as counter to expectations. However, from the perspective of extant research on the Big Two and conformity to sociocultural norms this negative relation is the expected one (if anything, the very small size of that relation is surprising) (for a review, see Gebauer, Lüdtke, Sedikides, & Neberich, 2014, JOPY)

We are aware of research conducted by Gebauer and colleagues. In fact, our study is based on Bakan’s expressiveness perspective that they apply and expand in their research. According to both Bakan’s perspective and Gebauer et al.’s research, as you suggested, it is the expression of communion and not agency that is allowed in the context of prosociality. In our theoretical reasoning we used that same starting point. However, taking into account the fact that studies on adolescents’ prosocial behavior (e.g., Eberly-Lewis & Coetzee, 2015), as well as context-specificity of pandemic, both suggest that socially responsible behavior in the COVID-19 pandemic can be driven by both intentions to protect others form the virus (communion) and intention to protect oneself from the virus (agency), we hypothesized that the context-specificity of our data collection allows, in fact, expression of both agency and communion. Moreover, studies within the same COVID-specific context showed an increase in self-oriented motives (agency) as a compensatory mechanism (Zhao et al., 2021), and indicated that participants mostly followed guidelines that protected themselves rather than the general public (Leder et al., 2021). Hence, positive relations of both agency and communion were expected in our study. This is why the negative relation between agency and norm adherence was described as counter to expectations.

We did, however, reflect more in detail on Gebauer et al.’s (2014) findings and connect it with ours in the discussion now (page 17):

“Similarly, this finding is, to some extent, in line with studies on prosocial behavior among adolescents [23], as well as among adults [22], that showed communion as a positive, but agency as a negative factor. Moreover, Gebauer et al. [22] found communion to be a stronger predictor of prosocial behavior in countries where interest in prosociality is common, and weaker in countries where such behavior is uncommon (communal social assimilation). In contrast, agency negatively predicted interest in prosocial behavior comparatively weakly in countries where interest in prosociality was common, and comparatively strongly in sociocultural contexts where such interest was uncommon (agentic social contrast). Applying that to our results, and insofar prosocial behavior can be equated to socially responsible behavior during the pandemic, this could indicate a stronger interest in prosociality in Austria where data was collected.”

Your comment also made us realize that we previously cited wrong Gebauer et al.’s study in our manuscript. That is corrected now in the reference list (reference No. 22, page 24).

References mentioned in the rebuttal letter “Response to Reviewers”: 

Eberly-Lewis, M. B., & Coetzee, T. M. (2015). Dimensionality in adolescent prosocial tendencies: Individual differences in serving others versus serving the self. Personality and Individual Differences, 82, 1–6. https://doi.org/10.1016/j.paid.2015.02.032

Gebauer, J. E., Paulhus, D. L., & Neberich, W. (2013). Big Two Personality and Religiosity Across Cultures: Communals as Religious Conformists and Agentics as Religious Contrarians. Social Psychological and Personality Science, 4(1), 21–30. https://doi.org/10.1177/1948550612442553

Gebauer, J. E., Sedikides, C., Lüdtke, O., & Neberich, W. (2014). Agency-communion and interest in prosocial behavior: social motives for assimilation and contrast explain sociocultural inconsistencies. Journal of personality, 82(5), 452–466. https://doi.org/10.1111/jopy.12076

James, G., Witten, D., Hastie, T., & Tibshirani, R. (2013). An Introduction to Statistical Learning (Vol. 103). Springer. https://doi.org/10.1007/978-1-4614-7138-7

Johnston, R., Jones, K., & Manley, D. (2018). Confounding and collinearity in regression analysis: a cautionary tale and an alternative procedure, illustrated by studies of British voting behaviour. Quality & Quantity, 52(4), 1957–1976. https://doi.org/10.1007/s11135-017-0584-6

Leder, J., Pastukhov, A., & Schütz, A. (2020). Even prosocially oriented individuals save themselves first: Social Value Orientation, subjective effectiveness and the usage of protective measures during the COVID-19 pandemic in Germany. https://doi.org/10.31234/osf.io/nugcr

Zhao, L., Ding, X., & Yu, F. (2020). Public moral motivation during the COVID-19 pandemic: Analysis of posts on Chinese social media. Social Behavior and Personality: An international journal, 48(11), e9829.

---

## [Editor Report · Decision Letter 1]

13 May 2022

The role of the Big Two in socially responsible behavior during the COVID-19 pandemic: Agency and communion in adolescents’ personal norm and behavioral adherence to instituted measures

PONE-D-22-00800R1

Dear Dr. Korlat,

We’re pleased to inform you that your manuscript has been judged scientifically suitable for publication and will be formally accepted for publication once it meets all outstanding technical requirements.

Kind regards,

Peter Karl Jonason

Academic Editor

PLOS ONE
---

## [Editor Report · Acceptance letter]

1 Jun 2022

PONE-D-22-00800R1 

The role of the Big Two in socially responsible behavior during the COVID-19 pandemic: Agency and communion in adolescents’ personal norm and behavioral adherence to instituted measures 

Dear Dr. Korlat:

I'm pleased to inform you that your manuscript has been deemed suitable for publication in PLOS ONE. Congratulations! Your manuscript is now with our production department. 

Kind regards, 

on behalf of

Dr. Peter Karl Jonason 

Academic Editor

PLOS ONE